# Spontaneous symmetry breaking of dissipative optical solitons in a two-component Kerr resonator

Gang Xu [1,2], Alexander U. Nielsen[1,2], Bruno Garbin [1,2,3], Lewis Hill [4,5], Gian-Luca Oppo [4],
Julien Fatome[1,2,6], Stuart G. Murdoch [1,2], Stéphane Coen [1,2] & Miro Erkintalo [1,2 ✉]

Dissipative solitons are self-localized structures that can persist indefinitely in open systems driven out of equilibrium. They play a key role in photonics, underpinning technologies from mode-locked lasers to microresonator optical frequency combs. Here we report on experimental observations of spontaneous symmetry breaking of dissipative optical solitons. Our experiments are performed in a nonlinear optical ring resonator, where dissipative solitons arise in the form of persisting pulses of light known as Kerr cavity solitons. We engineer symmetry between two orthogonal polarization modes of the resonator and show that the solitons of the system can spontaneously break this symmetry, giving rise to two distinct but co-existing vectorial solitons with mirror-like, asymmetric polarization states. We also show that judiciously applied perturbations allow for deterministic switching between the two symmetry-broken dissipative soliton states. Our work delivers fundamental insights at the intersection of multi-mode nonlinear optical resonators, dissipative structures, and spontaneous symmetry breaking, and expands upon our understanding of dissipative solitons in coherently driven Kerr resonators.

[1] Department of Physics, University of Auckland, Auckland, New Zealand. [2] The Dodd-Walls Centre for Photonic and Quantum Technologies, Auckland, New Zealand. [3] Centre de Nanosciences et de Nanotechnologies (C2N), CNRS, Université Paris-Saclay, Palaiseau, France. [4] SUPA and Department of Physics, University of Strathclyde, Glasgow G4 0NG, Scotland. [5] National Physical Laboratory, Teddington, UK. [6] ICB, UMR 6303 CNRS, Université Bourgogne-Franche-Comté, Dijon, France. ✉email: m.erkintalo@auckland.ac.nz

Temporal cavity solitons (CSs) are the dissipative optical solitons[1–3] of coherently-driven nonlinear resonators[4]. First observed in a macroscopic optical fiber ring resonator[5], and soon thereafter in a monolithic microresonator[6], they are persistent pulses of light whose remarkable characteristics have attracted attention across the divide of fundamental and applied photonics. On the one hand, temporal CSs have revealed themselves as ideal entities for the systematic investigation of fundamental dissipative soliton physics, permitting controlled experimental insights into a range of nonlinear dynamical phenomena[7–12]. On the other hand, they have also enabled—particularly through their key role in the generation of coherent microresonator Kerr frequency combs[13–15]—ground breaking advances across numerous applications, including all-optical information processing[16,17], telecommunications[18,19], optical frequency synthesis[20], detection of extra-solar planets[21,22], spectroscopy[23,24] and ultrafast optical ranging[25–27].

Temporal CSs have hitherto been predominantly studied in the context of single-component (scalar) systems involving a single (spatial and polarization) transverse mode family of the resonator. It is only very recently that researchers have begun to explore the realm of multi-mode (vectorial) systems[28–32]. In particular, asymmetric excitation of two distinct mode families has been shown to allow for the simultaneous emergence of two non-identical CSs[29,31], enabling a route for the generation of multiple frequency combs from a single device[29]. However, solitons supported under strongly asymmetric conditions are still effectively scalar, being (almost) entirely associated with one of the modes excited. Whilst vectorial solitons that rely on a symbiotic combination between two orthogonal components have been extensively studied in single-pass waveguide propagation[33–35] and fiber lasers[36–38], there has so far been only a handful of theoretical studies on such structures in the context of temporal CSs in passive resonators[28,32], analyzed in the presence of asymmetric parameters with no experimental observations achieved. Here, we report on the first experimental study of CSs in a two-mode system under conditions of symmetric excitation, and remarkably show that asymmetric and mirror-like vectorial CSs can arise via the ubiquitous phenomenon of spontaneous symmetry breaking (SSB). Moreover, we show that appropriate perturbations applied on the cavity driving field allow for one symmetry-broken CS state to be transformed into the other, thus providing strong evidence of the spontaneous origins of the symmetry breaking[39].

Our experiments are performed in a macroscopic fiber ring resonator, where the two orthogonal polarization modes of the system are judiciously engineered to be degenerate[40]. We theoretically and experimentally show that, even when the two modes are equally driven, SSB gives rise to two distinct but co-existing CS states with mirror-like, asymmetric polarization states. The soliton symmetry breaking occurs via the same incoherent, Kerr cross-coupling mechanism that was theoretically proposed several decades ago to give rise to SSB of counterpropagating[41,42] or cross-polarized[43] homogeneous continuous wave (cw) intracavity fields, and that has recently been observed in experiments[40,44–47]. Our results demonstrate for the first time that such SSB dynamics can also occur for ultrashort, temporally localized CS states. We note that the possibility of both temporal and polarization symmetry breaking in resonators that are synchronously driven with short pulses has also been numerically identified[48], yet no experimental demonstrations or direct links to CS physics have been presented.

To the best of our knowledge, our results comprise the first experimental observations of SSB of temporal CSs (and non-homogeneous states in general) in a two-component Kerr resonator. Moreover, whilst SSB has been previously identified

observed for vectorial solitons of conservative systems[49,50], and studied theoretically in the context of various dissipative systems[51–53], the results presented in our work represent the first direct experimental observation of SSB of dissipative solitons in any two-component physical system. As such, our work provides fundamental insights at the intersection of two widely investigated nonlinear phenomena, linking together the rich physics of (vectorial) dissipative solitons[54–57] and SSB.

## Results

**Theory of CS symmetry breaking**. We consider a passive, coherently-driven ring resonator with self-focussing Kerr nonlinearity (anomalous group-velocity dispersion) that exhibits two incoherently coupled eigenmodes [see Fig. 1a]. In the mean-field (good cavity) limit, the slowly-varying envelopes $E_{1,2}(t, \tau)$ of the two eigenmodes obey coupled Lugiato–Lefever equations[28,31,32,43,48,58]. In dimensionless form, the equations read:

$$\frac{\partial E_{1,2}}{\partial t} = \left[ -1 + i(|E_{1,2}|^2 + B|E_{2,1}|^2 - \Delta_{1,2}) + i\frac{\partial^2}{\partial \tau^2} \right] E_{1,2} + F_{1,2} \tag{1}$$

Here $t$ is a slow time that describes the field envelopes' evolution at the scale of the cavity photon lifetime, whilst $\tau$ is a corresponding fast time that describes the envelopes' temporal profile. The coupling coefficient $B$ describes the strength of the Kerr cross-phase interaction, $\Delta_{1,2}$ describe the detunings of the driving fields from the respective cavity resonances, and $F_{1,2}$ describe the amplitude components of the driving field along the two eigenmodes. Note that, in contrast to ref. [48], we consider cw driving such that $F_{1,2}$ are constant scalars. In what follows, we refer to the total driving intensity $X = |F_1|^2 + |F_2|^2$.

We consider the situation where $\Delta_1 = \Delta_2 = \Delta$ and $F_1 = F_2 = \sqrt{X/2}$, such that Eq. (1) exhibit a perfect symmetry with respect to the interchange of the two modes, $E_1 \rightleftharpoons E_2$. To probe the intracavity states accessible in experiments, we perform a numerical simulation where the cavity detuning $\Delta$ is linearly increased from small to large values as a function of slow time $t$ (see caption of Fig. 1 for parameters and Methods for further details). This simulation mimics the experimental procedure typically used to excite soliton frequency combs in Kerr microresonators by scanning the driving laser frequency across a single cavity resonance[6]. To be closer to experimental conditions, we add uncorrelated white noise on the driving terms $F_1$ and $F_2$ at each integration step. Figure 1b shows the total integrated energy of each intracavity polarization mode as a function of the detuning. The polarization modes initially carry identical intensities, $|E_1|^2 = |E_2|^2$, but as the detuning increases beyond $\Delta > 3.2$, the energies part, revealing the breaking of the interchange symmetry $E_1 \rightleftharpoons E_2$. With further increase of the detuning ($\Delta > 6.5$), the symmetry is restored.

The symmetry breaking observed in Fig. 1b occurs when the detuning lies within the so-called "soliton step"—a signature widely used to identify the presence of CSs in scalar systems (microresonators in particular)[6]. To gain more insights, Fig. 1c, d show the spatiotemporal evolution of the intracavity intensity along the two different cavity modes as the detuning is increased, whilst Fig. 1e shows the corresponding evolution of (half the) total intensity [Fig. 1f, g show snapshots at selected detunings as indicated]. As in scalar systems, the intracavity fields initially correspond to homogeneous cw states, but undergo a Turing-like modulation instability that results in the formation of dissipative patterns that fill the entire system. Despite their complex

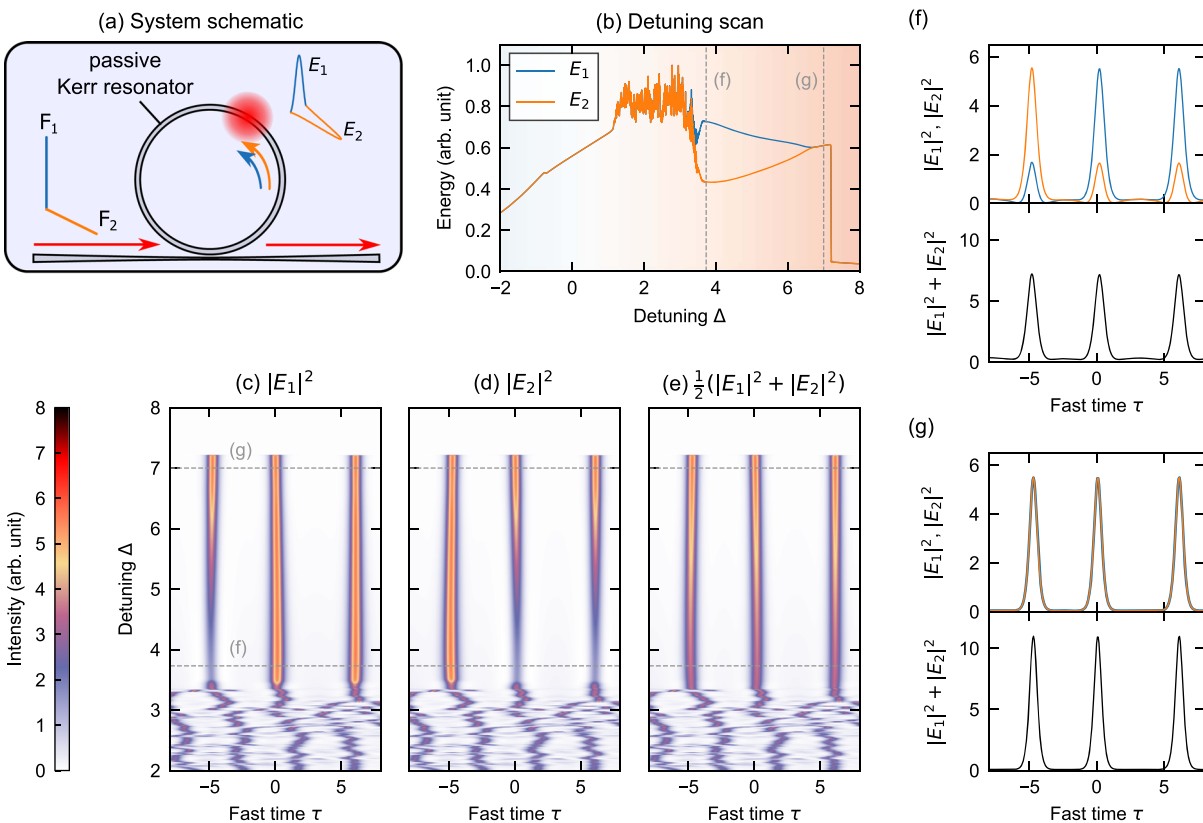

**Fig. 1 Concept and illustrative numerical simulations. a** Schematic illustration, showing a passive Kerr resonator whose two orthogonal (polarization) eigenmodes are symmetrically excited. **b** Numerical simulation results, showing the evolution of the integrated intracavity energy contained in cavity mode $E_1$ (blue curve) and $E_2$ (orange curve) as the detuning $\Delta$ is linearly increased. The energies part at the "soliton step", revealing SSB. **c**, **d** Evolution of the intracavity temporal intensity profiles corresponding to modes $E_1$ and $E_2$, respectively, as the detuning is changed; **e** shows the corresponding evolution of the total intensity. **f**, **g** Top panels show snapshots of the modal intensity profiles at detunings indicated with gray dashed lines in (**b**)–(**e**), whilst the bottom panels show the corresponding total intensity. All the results shown were extracted from the same numerical simulation with total driving intensity $X = 4.5$ and cross-coupling coefficient $B = 1.6$ similar to the experiments that follow. The colorbar represents the full modal intensities in (**c**) and (**d**) and half of the total intensity in (**e**).

dynamics, we find that the patterned states are (predominantly) identical across the two modes, $|E_1|^2 = |E_2|^2$.

When the detuning increases beyond $\Delta > 3.2$, localized CSs emerge from the patterned state. Remarkably, whilst the solitons that emerge carry identical total intensity [Fig. 1e], they come in two distinct flavors with imbalanced ($|E_1|^2 \neq |E_2|^2$) modal intensities, corresponding to mirror-like polarization states [Fig. 1c, d, f]. (Note that the splitting of the integrated energies witnessed in Fig. 1b arises due to there being a different number of solitons spontaneously excited with a dominant component along one of the modes.) As the detuning increases further beyond $\Delta > 6.5$, the symmetry of the solitons is recovered [Fig. 1g].

To better understand the emergence of asymmetric CS states, we computed the steady-state solutions of Eq. (1) using a Newton-Raphson relaxation algorithm[59] (see also Supplementary Note 1 for different parameter configurations). As shown in Fig. 2a, whilst symmetric CS solutions exist over a wide range of detunings, they are stable only for comparatively large detunings (the soliton stability was inferred based on a linearization analysis, see Methods). When the detuning decreases, the symmetric soliton loses its stability via a pitchfork bifurcation [P-CS in Fig. 2a], concomitant with the emergence of two stable states with asymmetric, mirror-like modal intensities. As is the case for the standard scalar CSs of single-mode systems[60], the symmetry-broken CS states exists down to the up-switching point

below which the homogeneous state no longer exhibits bistability (here $\Delta = 3.15$). At that point, the asymmetric soliton branches connect with branches describing correspondingly asymmetric patterned states. The patterned states originate from the homogeneous state via modulation (or Turing) instability at low detunings (not shown), and whilst they are initially symmetric, their intense peak intensity causes them to undergo their own pitchfork bifurcation [P-MI in Fig. 2a] as the detuning is increased. It is worth noting that the close proximity of the intersection points between the CS and HSS branches seen in Fig. 2a appears to be a peculiarity specific to the parameters used (see Supplementary Note 1 for bifurcation diagrams obtained for other driving powers).

The bifurcation curves shown in Fig. 2a suggest that, in close analogy with the well-known existence mechanism that underpins scalar CSs, the symmetry-broken vector CSs are underlain by the coexistence between a symmetry-broken patterned state and a stable, symmetric homogeneous background. We indeed find that the intensity profile of a symmetry-broken CS follows very closely a single period of a symmetry-broken patterned state that exists for the same parameters [see Fig. 2b, c]. In this context, we must emphasize that the soliton symmetry breaking does not require a simultaneous breaking of the (corresponding) symmetry of the homogeneous state; results in Figs. 1 and 2 were in fact obtained using a driving power that is below the threshold of SSB of the homogeneous state[42,48].

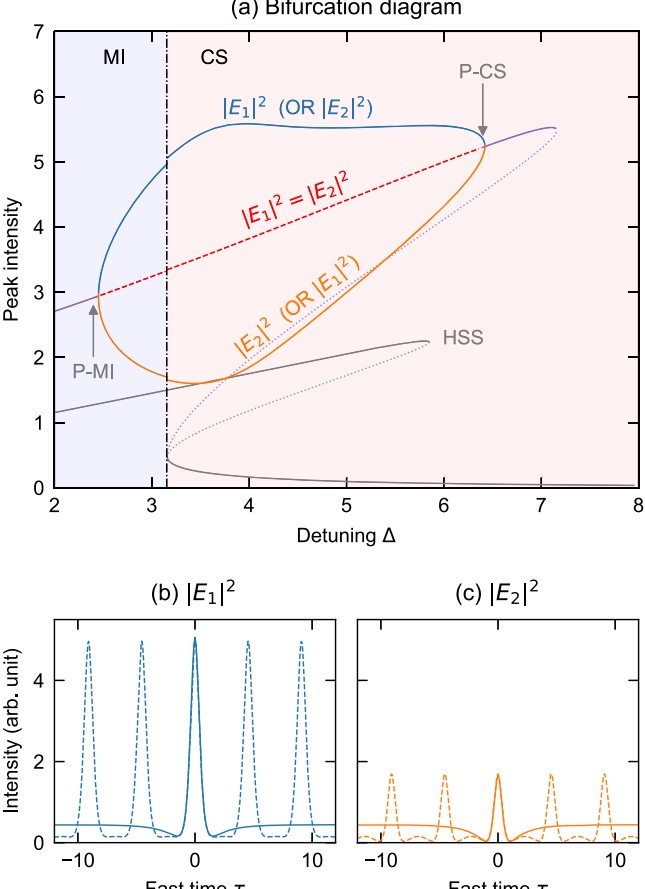

**Fig. 2 Bifurcations of symmetry-broken CSs. a** Bifurcation diagram showing the intensity of the homogeneous steady-state (HSS) solutions and the peak intensity of the CS and selected modulation instability (MI) pattern solutions of Eq. (1) as indicated [parameters as in Fig. 1]. The HSS solutions are symmetric at all detunings, whilst the CS and patterned solutions undergo a symmetry breaking pitchfork bifurcation at detunings 6.4 (P-CS) and 2.4 (P-MI), respectively. Solutions that are unstable against polarization symmetry breaking are indicated with red dashed lines. The unconditionally unstable CS and HSS branches are indicated by dotted lines. **b** and **c** Compare the modal intensity profiles for a symmetry-broken CS (solid curves) and a patterned solution (dashed curves) that both exist just beyond the up-switching point at $\Delta = 3.16$.

**Experimental observations**. For experimental demonstration, we use a setup that is similar to the one used in ref. [40] to observe the symmetry breaking of homogeneous states [see also Methods]. There is, however, one key difference: whereas the resonator used in ref. [40] exhibited normal dispersion to avoid CSs or related pattern forming instabilities, our setup uses a resonator made out of standard single-mode optical fiber (SMF-28) with anomalous dispersion $\beta_2 = -20$ ps$^2$/km at the driving wavelength of 1550 nm.

We synchronously drive the resonator with flat-top, quasi-cw pulses with 4.5 ns duration carved from a narrow-linewidth distributed feedback (DFB) cw fiber laser centred at 1550 nm. To overcome small residual desynchronization between the resonator and the nanosecond driving pulses, we imprint a shallow phase modulation at 2.87 GHz atop the driving field. This phase modulation acts as an attractive potential that traps the solitons[16], but we have carefully verified that it does not interfere with the SSB phenomenon.

The eigenmodes of interest in our experiments correspond to the two orthogonal polarization modes of the resonator—they are the principal states of polarization that return to their initial state after one complete round trip. We avoid linear mode coupling between the two modes (which would give rise to phase-sensitive interactions that complicate the cavity dynamics) by driving them with different carrier frequencies spaced by about 80 MHz. As detailed ref. [40], the carrier frequency shift is chosen to cancel the fiber birefringence such that $\Delta_1 = \Delta_2$; additional careful control on the pump levels maintains $F_1 = F_2$, allowing us to reliably reach symmetric operating conditions. We project the light output via a 99/1 intracavity coupler along the two polarization modes of the resonator by means of a polarizing beam splitter, and detect each component with both slow (10 kHz) and fast (12.5 GHz bandwidth) photodetectors; the former averages over several cavity round trips whilst the latter gives access to fast temporal dynamics over a single-round trip.

Figure 3a shows typical (slow) photodetector traces along each polarization component as the laser frequency is scanned over a single cavity resonance. Here the peak power of the quasi-cw driving field was set to 0.4 W, yielding a normalized driving amplitude $X = 4.5$ that is similar to the simulations above [c.f. Fig. 1]. In remarkable qualitative agreement with those simulations, we see clearly how the intensities of the two polarization components exhibit a "bubble" profile: the intensities are initially identical, part when the detuning approaches the soliton step, and again coalesce as the detuning increases further.

To demonstrate that the system supports two symmetry-broken CSs, we lock the detuning in the soliton step region ($\Delta = 3.9$) using the scheme described in ref. [61]. We then excite solitons by mechanically perturbing the resonator. Figure 3b depicts polarization-resolved oscilloscope traces measured using the fast 12.5 GHz photodetector. The intracavity field is clearly composed of two distinct types of localized structures with asymmetric, mirror-like polarization states. In stark contrast, repeating the measurement at a larger detuning $\Delta = 6.5$, we observe the CSs to be completely symmetric [Fig. 3c], in agreement with the theoretical predictions in Figs. 1 and 2.

The traces shown in Fig. 3b, c are limited by the bandwidth of our detection system, and therefore do not reflect the solitons' actual temporal profile. In Fig. 3d, we plot the polarization-resolved spectrum of a single symmetry-broken CS (at $\Delta = 3.9$) for which the $E_1$ mode dominates (the $E_2$ component is subdued by about 4.3 dB). The soliton spectra have sech$^2$ profiles with a 3 dB bandwidth of 0.11 THz along each mode, corresponding to a temporal pulse duration of 2.7 ps. This is in good agreement with the duration of 2.9 ps predicted by numerical simulations of Eq. (1) using our experimental parameters. We must emphasize that, while the spectrum shown in Fig. 3d is strictly representative of only one of the two soliton states ($E_1$ dominates), we can also readily observe the other soliton state ($E_2$ dominates) which displays identical characteristics: the only discernable difference between the two solitons is their state of polarization.

The observations in Fig. 3b, d demonstrate that, while the symmetry-broken solitons are predominantly polarized along one of the cavity modes, they also exhibit a noticeable component along the other mode: the solitons are vectorial [see also Supplementary Note 3]. In good qualitative agreement with the bifurcation curve shown in Fig. 2a, our experiments show that the contrast between the two components of the soliton changes with the cavity detuning [blue solid circles in Fig. 3e]. These results were obtained by locking the detuning at different values and by extracting the soliton's modal energies from the detected oscilloscope traces. (Note: as per well-known Kerr cavity dynamics[60], at detunings smaller than the range of soliton

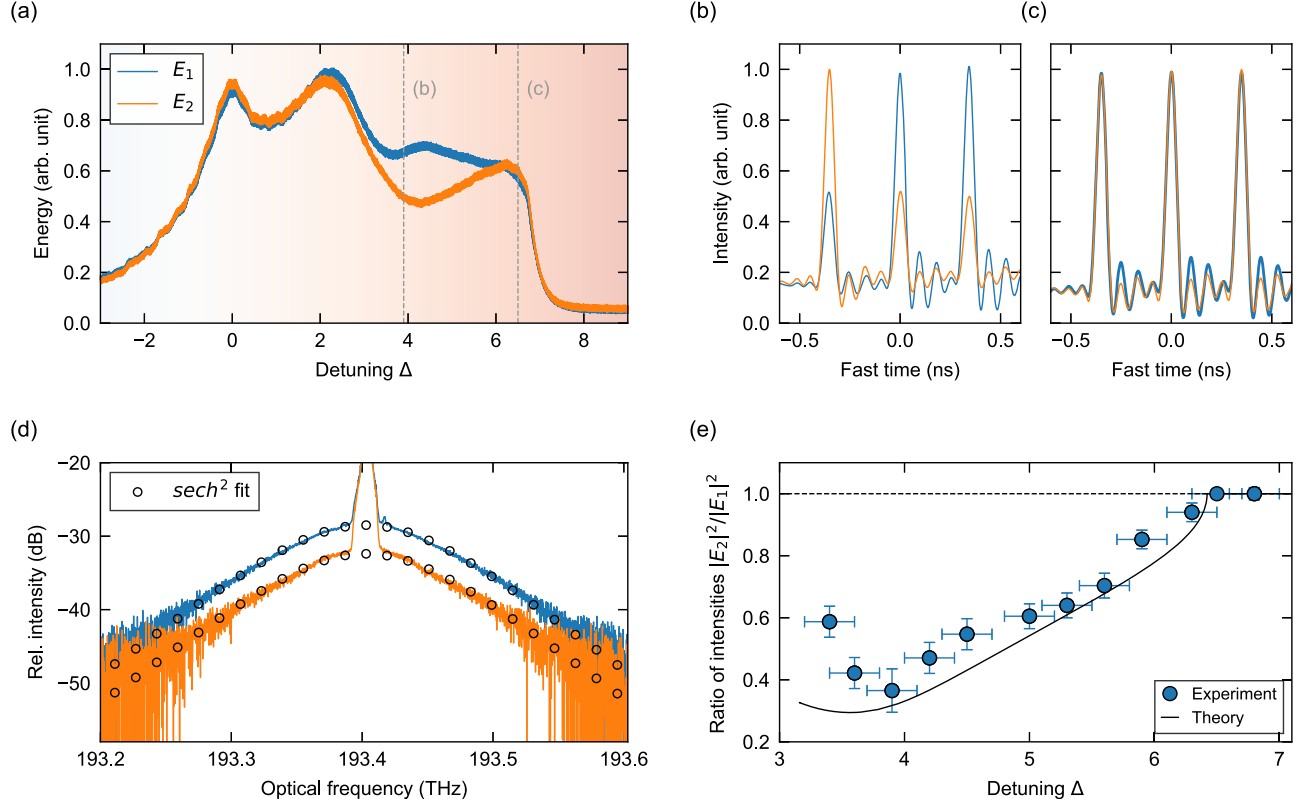

**Fig. 3 Experimental observation of CS symmetry breaking. a** Slow photodetector signals measured along the two cavity modes when scanning the laser frequency across a resonance. The experimental parameters are similar to the ones used to obtain the simulation results in Fig. 1: $X = 4.5$ and $B = 1.6$. **b** Intracavity intensity profiles measured with fast photodetectors (and subsequently sinc-interpolated), showing (**b**) asymmetric solitons at a detuning of $\Delta = 3.9$ and (**c**) symmetric solitons at a detuning of $\Delta = 6.5$. **d** Optical spectrum of one of the symmetry-broken CS states. The soliton is predominantly polarized along the "blue mode" ($E_1$), but exhibits a small component along the orthogonal "orange mode" ($E_2$). Also shown as open circles is the spectrum expected for a 2.7 ps hyperbolic secant CS. **e** Experimentally measured (blue solid circles) and theoretically predicted (black solid curve) ratio of the soliton intensity along the two polarization modes for a "blue mode" dominated CS as a function of detuning. The horizontal error bars represent an estimated ± 0.2 uncertainty in the detuning whilst the vertical error bars represent the standard deviation of multiple measurements (see Methods). The theoretical curve in (**e**) was obtained from the bifurcation data shown in Fig. 2a.

existence covered in Fig. 3e, the intracavity waveform corresponds to an unstable modulation instability pattern, whilst at larger detunings, the intracavity waveform corresponds to a homogeneous cw state.) As can be seen, the solitons are symmetric only at comparatively large detunings, become asymmetric as the detuning is decreased, and remain asymmetric until they cease to exist at $\Delta = 3.15$. Quantitatively, the observed ratios deviate from the numerical predictions—especially at low detunings—which we attribute to a range of experimental imperfections, including uncertainties in experimental parameters, the finite response time of our detection scheme, residual asymmetries, and higher-order effects not included in our model (e.g., stimulated Raman scattering and higher-order dispersion). Nonetheless, these results [together with the results shown in Fig. 3b, d] clearly confirm the salient theoretical predictions derived from Fig. 1 and 2: CSs can undergo SSB in a two-component Kerr resonator, giving rise to coexistence between two distinct, mirror-like vector soliton states. We must emphasize that, being based on a SSB phenomenon, these dynamics are fundamentally different compared to other scenarios where explicitly broken symmetries have been shown to lead to multiplexing of distinct quasi-scalar soliton states[9,29,31,62].

The soliton symmetry breaking demonstrated above is not limited to the particular parameters used, but occurs whenever the driving power $X$ exceeds a certain threshold (that depends on the cross-coupling coefficient $B$, see Supplementary Note 1). For

each of the driving powers that we have tested, it is relatively straightforward to adjust the experimental parameters to be sufficiently close to perfect symmetry to permit the signature features of the soliton symmetry breaking—a fact that attests to the robustness of the phenomenon. In Fig. 4a, we show photodetector traces measured as the laser frequency is scanned over a cavity resonance with the peak power of the driving pulses set to 1.7 W (corresponding to $X = 21$). The splitting of the modal intensities in the vicinity of the soliton step is evident. We also find that the contrast between the solitons' modal intensities increases with the driving power; for large $X$, the symmetry-broken solitons are aligned almost entirely along one of the cavity modes [see e.g., inset of Fig. 4a and Supplementary Notes 1 and 2].

The experimental observation of coexistence between two distinct soliton states with mirror-like polarization states [as shown, e.g., in Fig. 3b] provides compelling evidence of the spontaneous origins of the soliton symmetry breaking. To provide even stronger evidence, we eliminate any sources of explicit symmetry breaking that may arise from inhomogeneities of the cavity driving field, and show that it is possible to deterministically switch one soliton state to the other via appropriate perturbations. (Note that switching between two homogeneous states has previously been used to unequivocally demonstrate SSB in coupled photonic-crystal nanolasers[39].) We achieve this in our experiments by using a polarization modulator to locally perturb

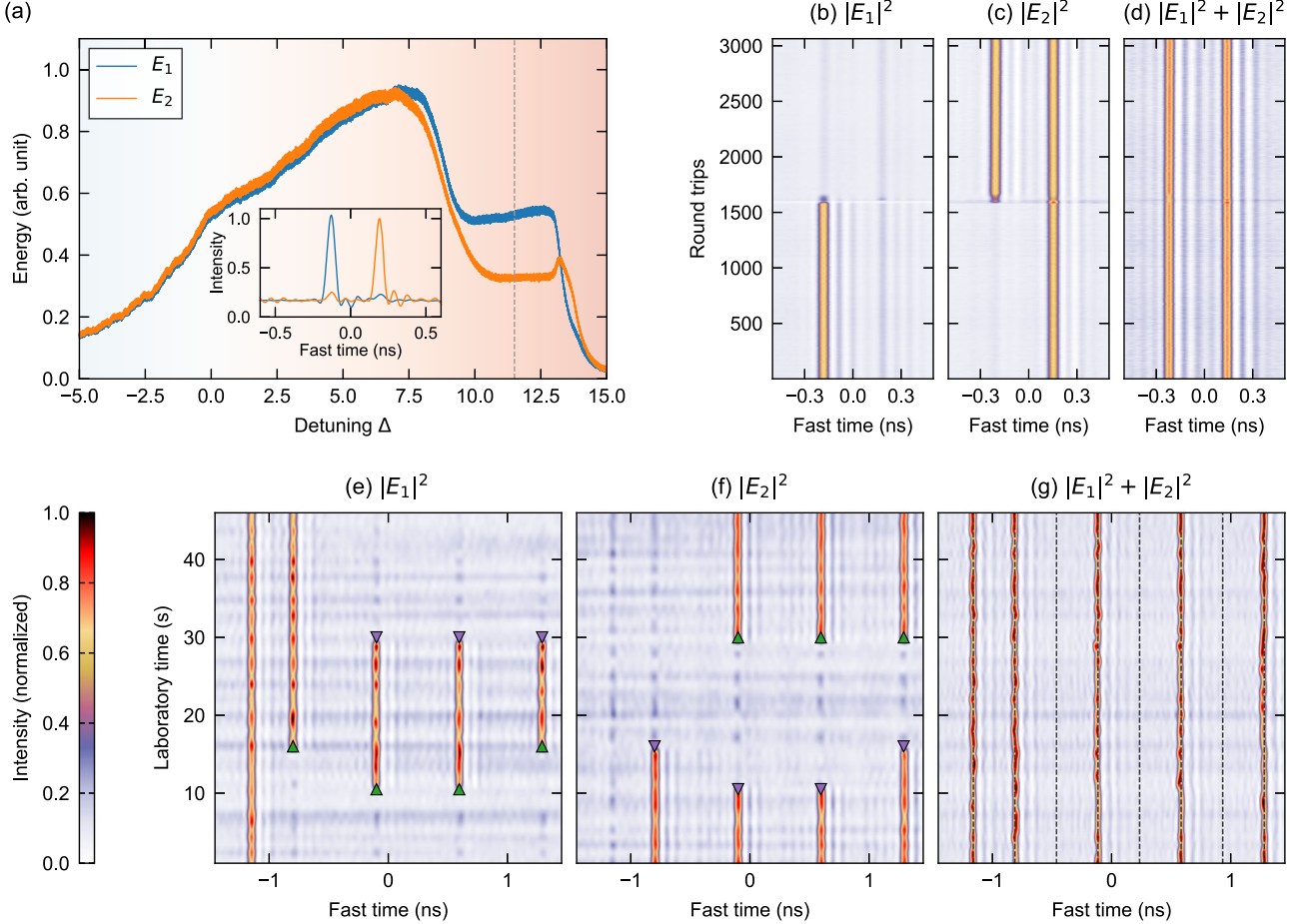

**Fig. 4 Observation and deterministic switching of symmetry-broken CSs for $X = 21$. a** Slow photodetector signals measured along the two cavity modes when scanning the laser frequency across a resonance. Inset shows an oscilloscope trace when the detuning is locked at $\Delta = 11.5$ (gray dashed vertical line). **b–d** Space-time diagrams, showing how two CSs with different polarization states evolve from roundtrip-to-roundtrip: (**b**) $E_1$ component, (**c**) $E_2$ component, (**d**) total intensity. A polarization perturbation is applied on the driving field at round trip 1600, and can be seen to enact deterministic switching of a CS from one polarization to the other. **e–g** Simultaneous and parallel polarization switching of several CSs: (**e**) $E_1$ component, (**f**) $E_2$ component, (**g**) total intensity. The green and magenta triangles highlight polarization increase and decrease perturbations, respectively. The colorbar applies to all (**b–g**) and the dashed vertical lines in (**g**) indicate the 2.87 GHz grid defined by the pump phase modulation.

the cavity driving field. Thanks to the timing reference provided by the phase modulation used to trap the solitons, we are able to selectively address, and hence switch, individual solitons. Our experiments show that a localized reduction in intensity along polarization mode $E_1$ (which corresponds to an increase along mode $E_2$) applied on the driving field at a position with a CS in the $E_1$ mode allows that soliton to be switched onto the other $E_2$ mode (and vice versa).

Figures 4b–d show real-time experimental measurements that demonstrate deterministic switching of soliton polarization [see also Supplementary Note 4]. Here we plot vertically concatenated sequences of fast oscilloscope traces measured along the two polarizations [Fig. 4b, c] as well as the corresponding total intensity [Fig. 4d]. The experiment is initialized with two solitons close to each other: the leading (left) soliton is (predominantly) aligned along mode $E_1$ while the trailing (right) soliton is (predominantly) aligned along mode $E_2$. From the start of round trip 1600, we apply a polarization perturbation on the cavity driving field that consists of an intensity decrease (increase) along the $F_1$ ($F_2$) component, and that is localized over a 2 ns temporal interval that encompasses both solitons (the perturbation is synchronously applied for about one cavity photon lifetime, or seven round trips). As can be seen, the perturbation causes the leading CS in the $E_1$ mode to switch into the $E_2$ mode. In stark

contrast, the perturbation has no effect on the trailing soliton, since an intensity decrease along $E_1$ (increase along $E_2$) does not permit the switch $E_2 \rightarrow E_1$. Likewise, we observe no lasting effect at temporal positions where a CS is not initially present. This observation demonstrates that the perturbation is not sufficiently strong to excite new solitons, but rather only allows for the switching of one mirror-like state to the other.

Several solitons can be switched individually and in parallel, allowing for complex all-optical manipulations of polarization-multiplexed information. In Fig. 4e–g, we demonstrate the manipulation of an 8-bit sequence at 2.87 GHz (defined by our phase modulation) that consists of five solitons (and three empty bit slots). Over the course of the measurement, several of the solitons are switched from one polarization to the other and back. Remarkably, the total intensity remains almost constant during the manipulations [see Fig. 4g], indicating pure polarization dynamics.

## Discussion

We have reported on the theoretical prediction and experimental observation of SSB of temporal CSs in a two-mode Kerr nonlinear ring resonator. The SSB instability gives rise to two co-existing CS states with asymmetric, mirror-like intensity distributions across the two modes of the system. We have obtained clear

experimental evidence of such soliton symmetry breaking, and shown that appropriate perturbations enable deterministic switching between the two asymmetric soliton states. Our experimental observations are in very good agreement with numerical simulations.

Whilst a detailed discussion is beyond the scope of the present Article, our experiments and simulations reveal that the symmetry breaking is not limited to stable CSs, but can also occur for periodically and chaotically oscillating solitons [see Supplementary Note 2]. Our work paves the way for further studies into the rich dynamics of SSB effects in two-component Kerr resonators. To the best of our knowledge, our results constitute the first direct experimental observations of SSB of vectorial dissipative solitons in any two-component physical system. As such, our work provides connections and insights at the interface of vector solitons, dissipative solitons, and SSB.

## Methods

**Additional theoretical details.** All the numerical calculations presented in our work are based on Eq. (1). The results in Fig. 1 were obtained using a split-step Fourier integration scheme with the nonlinear step evaluated using the fourth-order Runge–Kutta algorithm. The simulation linearly ramps the detuning from $\Delta = -3$ to $\Delta = 8$ over a slow time interval of $\Delta t = 1500$ in 1.5 million integration steps. We must emphasize that, since the symmetry breaking instability takes time to spontaneously develop from noise, the precise dynamics observed may depend on the rate with which the detuning is scanned. In the simulations shown in Fig. 1, we additionally add uncorrelated white noise with random phase and amplitude on both driving fields at each integration step so as to facilitate the onset of the symmetry breaking. Besides instigating the symmetry breaking, the noise does not perturb the salient CS dynamics [see e.g., Fig. 1e]. It is also worth emphasizing that no effort was made to match the noise used in the simulations with the noise present in our experiments, which (together with other technical issues and experimental uncertainties) is likely to explain the small discrepancies between the modelled and observed resonance scans.

The bifurcation diagrams and steady-state solutions shown in Fig. 2 were obtained by solving the roots of Eq. (1) by means of a multidimensional Newton-Raphson algorithm. The stability of the steady-state solutions was evaluated based on a standard linearization analysis. Specifically, we examine the eigenvalues of the Jacobian matrix of the right-hand-side of Eq. (1) with the fast time discretized on a uniform grid. There are in principle infinitely many patterned solutions with different periods (or frequencies) that can be found. The particular patterned solution displayed in Fig. 2 was chosen as a compromise that captures the salient details (e.g., SSB transition) whilst still connecting relatively smoothly (yet with a noticeable discontinuity) to the CS state.

The normalization of Eq. (1) is the same as the one used in studies of scalar CSs e.g., in ref. [5]. In particular, the normalized driving power referred to in our work is defined as

$$X = \frac{\gamma P_{\text{tot}} L \theta}{\pi^3} \mathcal{F}^3, \qquad (2)$$

where $\gamma$ is the Kerr nonlinearity coefficient, $P_{\text{tot}}$ is the total driving power, $L$ is the resonator length, $\theta$ is the power coupling coefficient used to launch the driving field into the resonator, and $\mathcal{F}$ is the cavity finesse. Parameters relevant to our experiments are quoted below or in the main manuscript above.

**Additional experimental details.** The fiber ring resonator used in our experiments is made of a 86 m-long segment of standard single-mode optical fiber (SMF-28). We estimate that the resonator exhibits a group-velocity dispersion coefficient $\beta_2 \approx -20\text{ps}^2\,\text{km}^{-1}$ and Kerr nonlinearity coefficient $\gamma \approx 1.2\text{W}^{-1}\text{km}^{-1}$ at the driving wavelength of 1550 nm. The resonator incorporates a 95/5 coupler to inject the coherent driving field into the cavity ($\theta = 0.05$) and a 99/1 coupler through which the intracavity dynamics can be monitored. The resonator also includes a fiber polarization controller that allows to control the resonator's net birefringence. We measured the resonator finesse to be $\mathcal{F} = 42 \pm 2$, corresponding to 58 kHz resonance linewidth.

The 4.5 ns flat-top, quasi-cw pulses used to drive our resonator are carved from a narrow-linewidth (<1 kHz) DFB fiber laser (Koheras Adjustik E15) using a Mach–Zehnder amplitude modulator that is driven by a pulse pattern generator (PPG). The PPG in turn is referenced to an external clock whose frequency (about 2.87 GHz) is carefully adjusted to be an integer multiple of the 2.39 MHz ± 5 kHz free-spectral range of the resonator. This ensures the driving pulses are synchronous with the 420 ns cavity round trip time. The nanosecond pulses then pass through a phase modulator that is driven by the same clock signal that references the PPG. The resulting 2.87 GHz sinusoidal phase profile acts as a trapping potential for the CSs, allowing for the mitigation of small residual desynchronization between the driving pulses and the cavity round trip time.

To symmetrically drive the two polarization modes of the resonator, we follow the method detailed in ref. [40]. Briefly, the driving pulses are first divided into two orthogonal components using a polarizing beam splitter, and one of the components is frequency shifted by about 80 MHz using an acousto-optic modulator (AOM). The two beams are then re-combined with a second polarizing beam splitter, and the polarization of the combined beam adjusted such that both resonator modes are equally driven (a feedback loop is used to maintain the power balance between the two modes to ensure long-term stability). The AOM frequency is then fine-tuned such that the two polarization components are simultaneously resonant, ensuring that the fiber's net birefringence is cancelled and the modal detunings always identical. After the total driving field is prepared in this manner, it is amplified with an Erbium-doped fiber amplifier and spectrally filtered to remove amplified spontaneous emission before being launched into the resonator through the 95/5 coupler.

The experimental results shown in Figs. 3a and 4a were obtained by slowly ramping the frequency of the driving DFB laser across a single cavity resonance. Note that the splitting (or merging) of the modal intensities observed in these measurements do not display a clean square-root dependence characteristic of pitchfork bifurcations [see Fig. 2a] because: (i) the laser frequency (hence, cavity detuning $\Delta$) is continuously increased such that the SSB does not have time to adiabatically develop (or disappear) close to the bifurcation points and (ii) the slow photodetector used to record the data has a response time corresponding to about 250 round trips. Experimental results other than those shown in Figs. 3a and 4a were obtained with the pump laser frequency fixed at a constant frequency detuning from the closest cavity resonance. This was achieved by using the detuning stabilization scheme first introduced in ref. [61]. Specifically, a low-power signal derived from the DFB laser is frequency shifted with a second AOM and launched into the resonator such that it counterpropagates with respect to the main pumps. The intracavity power level of this signal is then locked to a set value using a proportional-integral-derivative controller, thus actively stabilizing the detuning. By controlling this AOM frequency shift, we are able to systematically adjust the values of the cavity detunings. Specifically, the value of the normalized detuning is given by $\Delta \approx 2(f_0 - f_R)\mathcal{F} / \text{FSR}$, where $f_0$ is the frequency applied on the AOM and $f_R = 79.04\text{MHz} \pm 6\text{kHz}$ is a reference frequency that corresponds to zero detuning. The latter reference frequency is deduced by overlapping the backward resonance with the conspicuous first peak of the forward resonance [see e.g., Fig. 3a] that arises from the residual cw background of the driving field. We estimate the error in the normalized detuning measured in this manner to be ± 0.2 [the horizontal error bars in Fig. 3e reflect this error].

The CS temporal traces shown in Figs. 3 and 4 were recorded with a 12.5 GHz photodetector, digitized with a 40 GSa/s oscilloscope, and interpolated with a sinc function. The intensity ratios shown in Fig. 3e were obtained by exciting multiple solitons in the cavity and then recording their intensity profiles for 3064 consecutive round trips (limited by the memory depth of our oscilloscope). For each detuning, this measurement was repeated several times and the average calculated over all solitons, over all round trips, and over all measurement realizations. (The vertical error bars describe the statistical spread in the entire data set.)

Data in Fig. 4b–d was obtained from the real-time acquisition of the oscilloscope over 3064 consecutive round trips. The resulting long time traces where chopped into single-round-trip segments, and concatenated above each other to build the spatiotemporal diagrams shown. The manipulations in Fig. 4e–g occur over much longer time scales, and so cannot be acquired in real time. These results were rather obtained by recording the oscilloscope traces once per second over a fixed time window that encompasses the nanosecond intracavity pump pulse.

## Data availability

The data that support the plots within this paper and other findings of this study are available from the corresponding author upon reasonable request.

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

## Acknowledgements

We acknowledge financial support from the Marsden Fund, the Rutherford Discovery Fellowships, and the James Cook Fellowships of the Royal Society of New Zealand. J.F. thanks the financial support from the CNRS through the IRP Wall-IN project. L.H. thanks the financial support from the Mac Robertson Trust, the Dodd Walls Centre, and EPSRC DTA through Grant No. EP/M506643/1

## Author contributions

G.X. performed all the experiments reported in this paper based on a setup developed by A.U.N, who also made the first observations of the CS symmetry breaking phenomenon with the help of B.G. Overall, G.X. and A.U.N. contributed equally to this work. B.G., G.L.O and M.E. came up with the original idea. The experimental setup is built on two earlier setups constructed by B.G., S.C. and A.U.N. G.L.O, L.H. and J.F. contributed to the interpretation of the results and provided theoretical assistance. S.G.M. provided support for the experimental work and together with S.C. and M.E. supervised the project. S.C. computed the results shown in Fig. 2 and together with M.E. obtained funding for the project. M.E. additionally performed the numerical simulations in Fig. 1 and wrote the paper with feedback from all the authors.

## Competing interests

The authors declare no competing interests.
