## [Peer Review File · Nature Communications]

Reviewers' Comments:

Reviewer #1:

Remarks to the Author:

The authors have done good work addressing criticism and implementing suggestions of the reviewers. The paper has gained sufficient clarity and is suitable for publication in the present form.

Reviewer #3:

Remarks to the Author:

Compared with the previous version of this paper, the main change is the more thorough theoretical investigation, which is unfortunately almost entirely relegated to the supplement. It establishes that SSB indeed most likely occurs for all $B > 1$, and that the broken symmetry solitons have a nontrivial polarization structure. On the other hand, it also demonstrates more clearly the discrepancy between the calculations and experiments: In the $X=21$ calculations the SSB takes place for breathers rather than stationary soliton that are observed experimentally. Of course, it is pointless to expect exact correspondence between theory and experiment, but here the key result - the SSB pitchfork bifurcation - is not faithfully reproduced by experiments. This mismatch raises the possibility that some important aspects the experimental SSB mechanism have not been understood. The claim that the inclusion of the smoothly varying cw power in the experimental measurements can smear the square root singularity is just wrong.

A possibly related issue is the sensitivity of the SSB phenomenon to deviations from perfect symmetry, which is not investigated either theoretically or experimentally. The authors answer to this question raised in my previous report is unsatisfactory. Is it possible that some aspects of the symmetry-conditions-stabilization mechanism in the experiments complicate the theoretical picture and explain the discrepancy between calculations and experiments?

Finally, regarding the question of the novelty of the results: This work is an impressive experimental achievement. On the other hand, conceptually it doesn't bring that much compared to what is already known, as the other referees of the previous version of this paper seem to think as well. Various aspects of the dynamical phenomenon studied here have been studied in so many previous works, including by the same group of authors, that it takes an effort to identify the precise points of novelty.

In summary, the authors have partially addressed the deficiencies pointed out in the previous report, but some problems remain. My view that this work represents a step forward but not a breakthrough hasn't changed.

Reviewer #4:

Remarks to the Author:

I have reviewed the paper "Spontaneous symmetry breaking of dissipative optical solitons in a two-component Kerr resonator by Xu and co-workers.

I generally agree with Referee 1, 2 and 3 that the experimental results are of very high quality and novel. Although this is a universal phenomenon, still the experimental demonstration in such optical setting provides important advancement in the general understanding. The authors have made a number of central improvements that clarified previous concerns.

The scientific innovation in my opinion is, at this point, well suited for Nature Communications, as symmetry breaking on dissipative solitons with different type of polarisation has never been observed before and highlights a number of important potentialities for dissipative systems in general.

I have been asked to revive the comments of the authors to the concerns of referee2. In my opinion, all these points have been fully cleared.

-Comment 1: regarding the appropriateness of the numerical methods, the authors are right, time dependent simulations and the search of stationary states are two different kind of approaches. Referee #2 compares time varying propagation and the linear stability analysis. Both approaches are needed for a full theoretical study. The second allows to understand the general stability of the stationary states, but for understanding how the system evolves this method is not sufficient and in this case temporal propagation is needed. I find that the authors are using the state-of-the-art numerical techniques for the problem.

-Comment 2: the study on polarisation effects in fibers needs indeed to be put into context. The first observation of fiber baser cavity solitons are from 2009 from some of the authors, so I clearly disagree that these effects were observed in the 80s. We do not need to confuse conservative and dissipative literature and, in the latter case, driven solitons from laser solitons. All these systems have a deeply different physics. I think that, however, this point has been fairly raised and that the clarification of the authors allows to put their work into a better context now.

-Comment 3: I find this a rather minor point.

That cavity solitons have been centrally studied as optical memories is undeniable (Barland et al. 'Cavity solitons as pixels in semiconductor microcavities' Nature volume 419, pages699-702(2002)), there is almost 30 years of literature.

That temporal cavity solitons are more useful for metrology applications in microcombs than as memories is, indeed, in my view, a matter of opinion, as Referee 2 also concedes. Given this comment, I think that the authors have well explained that , because the basic physics is the same, this work provides critical advancement for both fields, so I would say that also this concern has been cleared.

Referee 1: general remarks

The authors have done good work addressing criticism and implementing suggestions of the reviewers. The paper has gained sufficient clarity and is suitable for publication in the present form.

Referee 1: general response

We thank the Referee for their kind words. As they present no additional comments or criticism, we refrain from further response.

Referee 3: general remarks

Compared with the previous version of this paper, the main change is the more thorough theoretical investigation, which is unfortunately almost entirely relegated to the supplement. It establishes that SSB indeed most likely occurs for all $B > 1$, and that the broken symmetry solitons have a nontrivial polarization structure.

Referee 3: general response

We thank the Referee for carefully reading through our revised manuscript and for acknowledging the more thorough theoretical investigations presented. Please note that we decided to include the new theoretical investigations in the Supplement so as to keep the main manuscript succinct and focussed on the main results, namely experimental observations of spontaneous symmetry breaking of dissipative optical solitons. We genuinely believe that the new theoretical investigations provide valuable support for the main results, but that their specificity (and to a lesser extent complexity) could hinder the readability of the main manuscript and thus derail from the main messages that we wish to convey. We are also mindful of the fact that Referees 1 and 4 are both satisfied with the presentation of the manuscript in its present form, which makes us reluctant to shift any of the results from the Supplement to the main manuscript in fear of making the manuscript less accessible and readable.

Comment 1

On the other hand, it [new theoretical investigations] also demonstrates more clearly the discrepancy between the calculations and experiments: In the $X = 21$ calculations the SSB takes place for breathers rather than stationary soliton that are observed experimentally. Of course, it is pointless to expect exact correspondence between theory and experiment, but here the key result – the SSB pitchfork bifurcation – is not faithfully reproduced by experiments. This mismatch raises the possibility that some important aspects the experimental SSB mechanism have not been understood.

Response 1

The discrepancy referred to by the Referee in fact arises due to the detection scheme used in our experiments. Specifically, as the detuning is scanned over a resonance [see Figs. 2(a) and 4(a) of our main manuscript], we record the average intracavity modal intensities with slow photodetectors that average over several (approximately 250) round trips. This naturally smooths away oscillatory features that occur over timescales of tens of round trips (or less). The oscillation period of the breathers manifesting themselves in our simulations is about 10 round trips. As a consequence, it is not possible to resolve such oscillation in our detuning-scan experiments since the detector averages over about 250 round trips. Figure A below indeed illustrates how a response function with a duration of about 250 round trips applied on the numerically simulated data shown

Figure A: Simulated modal energies as the detuning is scanned. (a) Shows the raw simulation data [same data is shown in Fig. S3(a) of our Supplementary Information] whilst (b) shows the same data after it has been convolved with a response function that mimics the slow detector used in our experiments (response time of about 250 round trips). Note how oscillatory characteristics of breather solitons are smoothed out by the convolution. Also note that the increase of the convolved modal intensities towards the end of the scan in (b) is an artefact arising from the use of periodic boundary conditions when computing the convolution.

in Fig. S3(a) of our Supplementary Information smooths away oscillatory signatures of breather solitons. Of course, we must again emphasise that the exact profile of the intensity trace observed depends on the rate with which the detuning is scanned, as well as the specific noise present in the system.

Whilst signatures of breather solitons are not captured in our detuning-scan experiments (due to the averaging effect of the detector as mentioned above), we may expect such signatures if the detuning is actively stabilized in a regime where breathing solitons should exist. Remarkably, this hypothesis is fully confirmed in our experiments. Figure B below shows results from experimental measurements obtained for normalized driving intensity $X = 21$ when the normalized cavity detuning is stabilized at $\Delta = 10.5$ [all other experimental parameters as in Fig. 4 of our main manuscript]. We see clearly how the intracavity field consists of symmetry-broken solitons that exhibit periodic oscillations, as predicted by theory!

Taken together, the considerations above hopefully highlight that the $X = 21$ calculations *do not* reveal any fundamental discrepancies with our experiments. The experiments and the simulations as presented in our work show very good qualitative agreement. In fact, we continue to find it remarkable how such a simple model is capable of capturing the salient details of a complex experiment. **For the sake of completeness, we have decided to include the new experimental result shown in Fig. B as Fig. S4 in the Supplement, and we have also included a discussion on why the soliton breathers are not resolved in the time-traces measured with a slow photodetector.**

Comment 2

The claim that the inclusion of the smoothly varying cw power in the experimental measurements can smear the square root singularity is just wrong.

Response 2

We would like to thank the Referee for highlighting this issue. In hindsight, we communicated this point quite poorly. The square root dependence is not to be expected in our experiments predominantly because: (i) the detuning is continuously (and comparatively rapidly) scanned and (ii) the slow photodetector averages over several (about 250) round trips. (The averaging smooths out signatures of oscillatory CSs, as described above, but it also conspires with the continuously changing detuning to blur any square-root dependence). Because the

Figure B: Experimental results obtained for $X \approx 21$ and with the detuning actively stabilized at $\Delta \approx 10.5$, showing the roundtrip-by-roundtrip evolution of an oscillating, symmetry-broken CS. Panels (a) and (b) show modal intensities whilst (c) shows the corresponding total intensity. Panel (d) shows the polarization-resolved evolution of the integrated energy of a single CS, with blue and orange curves showing the energy along the E_1 and E_2 modes, respectively. Data in (d) was obtained by extracting the peak photodetector signal corresponding to the soliton at around -0.4 ns in panels (a)–(c).

Note: Figure added as new Fig. S4 in the Supplement.

SSB takes time to develop (or retreat), an experiment where the detuning is continuously scanned unavoidably smooths the bifurcation point; with regards to SSB, the detuning is not scanned adiabatically. On the other hand, it is technically quite challenging to scan the detuning any slower, as mechanical and thermal perturbations will give rise to random shifts of the cavity resonances over longer timescales. The slow response time of the detector further exacerbates the problem [see Fig. A]. **We have now revised our manuscript to highlight this point.**

Comment 3

A possibly related issue is the sensitivity of the SSB phenomenon to deviations from perfect symmetry, which is not investigated either theoretically or experimentally. The authors answer to this question raised in my previous report is unsatisfactory. Is it possible that some aspects of the symmetry-conditions-stabilization mechanism in the experiments complicate the theoretical picture and explain the discrepancy between calculations and experiments?

Response 3

To address the issue raised by the Referee, we have now theoretically investigated the sensitivity of the soliton SSB phenomenon to deviations from perfect symmetry. Specifically, applying a methodology as in ref. 40 of our manuscript, we have found the steady-state, symmetry-broken CS solutions in the presence of asymmetries in the driving field intensities and cavity detunings. For these calculations, we used a fixed total driving intensity $X = 4.5$ (as in Figs. 1–3 of our manuscript) and mean detuning $\Delta_m = (\Delta_2 + \Delta_1)/2 = 3.9$. Figure C shows results from our computations, with Fig. C(a) and C(b) showing the impact of asymmetric driving intensities and cavity detunings, respectively. Here, the driving field ellipticity angle defines the modal driving amplitudes viz. $F_1 = \sqrt{X} \cos \chi$ and $F_2 = \sqrt{X} \sin \chi$, whilst the difference in detuning $\delta\Delta = \Delta_2 - \Delta_1$, where $\Delta_2 = \Delta_m + \delta\Delta/2$ and $\Delta_1 = \Delta_m - \delta\Delta/2$.

As can be seen, the coexistence of two asymmetric soliton states is possible over finite ranges of asymmetries. Whilst these ranges are not very large, they are readily accessible with our experiments. Of course, as is

Figure C: Results from numerical continuation calculations that show the impact of asymmetries on the co-existing, vectorial CS states. Blue and orange colours indicate CS peak intensity along the E_1 and E_2 modes, respectively, with solid (dotted) curves corresponding to stable (unstable) states. In (a) the cavity detunings are constant (and symmetric) but the driving intensities are made asymmetric by varying the ellipticity χ . In (b), the driving intensities are kept constant (and symmetric) but the cavity detunings are made asymmetric around a mean value of $\Delta_m = 3.9$. In both cases, the total driving intensity $X = 4.5$. **Note:** Figure added as new Fig. S7 in the Supplement.

characteristic to SSB phenomena, fully random selection between the two different asymmetric states occurs only in the absence of any asymmetry (such that the pitchfork is whole). In the presence of asymmetries, one of the states dominates over the other.

In our experiments, we adjust the parameters to be as close as possible to being symmetric. This is achieved by meticulously looking for the splitting of the modal intensities when scanning over a resonance [e.g. Fig. 3(a) and 4(a) of our manuscript]. If the parameters are intentionally made asymmetric, the symmetry breaking vanishes. We believe that, at our optimal operating point, the parameters are sufficiently close to being symmetric such that the two coexisting soliton states (i) co-exist and (ii) can be spontaneously excited by the noise present in the system with almost equal probability. We do not believe that the active stabilization of the detuning or the driving field ellipticity has any impact on the actual symmetry breaking phenomenon. In fact, both of these stabilization schemes operate over comparatively slow (millisecond) timescales and only correct long-term drifts; they do not influence the nonlinear cavity dynamics that occur over the (microsecond) scale of the cavity photon lifetime. The symmetry breaking phenomenon is fragile in essence, yet sufficiently robust so as to be experimentally accessible.

We have now included the results shown in Fig. C as new Fig. S7 in our Supplementary Information document, accompanied by pertinent discussion.

Comment 4

Finally, regarding the question of the novelty of the results: This work is an impressive experimental achievement. On the other hand, conceptually it doesn't bring that much compared to what is already known, as the other referees of the previous version of this paper seem to think as well. Various aspects of the dynamical phenomenon studied here have been studied in so many previous works, including by the same group of authors, that it takes an effort to identify the precise points of novelty.

Response 4

We are grateful for the Referee's kind words regarding our experimental achievements. Unfortunately, we feel that it is misleading to state that various aspects of the dynamical phenomena have been studied before. In fact, the key dynamical phenomenon – observation of spontaneous symmetry breaking of dissipative (optical) solitons in a two-component system – has not to the best of our knowledge been studied in any previous work. We do not deny that our work builds upon previous discoveries and studies: this is the nature of all new science. However, we report on a new phenomenon that has never before been studied in experiment or theory, thus adding to the broad body of knowledge in the fields of dissipative solitons, driven resonators, and symmetry breaking. To quote Referee 4, "Although this is a universal phenomenon, still the experimental demonstration in such optical setting provides important advancement in the general understanding." and "...symmetry breaking on dissipative solitons with different type of polarisation has never been observed before and highlights a number of important potentialities for dissipative systems in general."

We also regret that the Referee finds that it takes an effort to identify the precise points of novelty. In fact, in the abstract of our manuscript, on lines 55-64 of our manuscript, on lines 86-96 of our manuscript, and on lines 442-445 of our manuscript, we explicitly state how our results comprise the *first experimental observations* of spontaneous symmetry breaking of dissipative solitons in a two-component system (amongst other things). We feel these statements make the precise points of novelty quite clear, and we are not entirely sure how to make them clearer. Lastly, with regards to opinions of other Referees (as alluded to in the comment), we would like to respectfully note that Referees 1 and 4 consider our paper ready to be published in Nature Communications.

Comment 5

In summary, the authors have partially addressed the deficiencies pointed out in the previous report, but some problems remain. My view that this work represents a step forward but not a breakthrough hasn't changed.

Response 5

We would like to thank the Referee for their feedback. We hope that the clarifications above and the further changes made to our manuscript address the remaining problems. We acknowledge that what constitutes a breakthrough is subjective, especially at the point of dissemination of any scientific progress. We believe that the quality, novelty, and scope of our work is suitable for Nature Communications, and we also believe that the constructive comments from all of the Referees have been very helpful in shaping our manuscript (and Supplement) to an optimal form.

Referee 4: general remarks

I have reviewed the paper "Spontaneous symmetry breaking of dissipative optical solitons in a two-component Kerr resonator" by Xu and co-workers.

I generally agree with Referee 1, 2 and 3 that the experimental results are of very high quality and novel. Although this is a universal phenomenon, still the experimental demonstration in such optical setting provides important advancement in the general understanding. The authors have made a number of central improvements that clarified previous concerns.

The scientific innovation in my opinion is, at this point, well suited for Nature Communications, as symmetry breaking on dissipative solitons with different type of polarisation has never been observed before and highlights a number of important potentialities for dissipative systems in general.

I have been asked to revive the comments of the authors to the concerns of referee 2. In my opinion, all

these points have been fully cleared.

- Comment 1: regarding the appropriateness of the numerical methods, the authors are right, time dependent simulations and the search of stationary states are two different kind of approaches. Referee #2 compares time varying propagation and the linear stability analysis. Both approaches are needed for a full theoretical study. The second allows to understand the general stability of the stationary states, but for understanding how the system evolves this method is not sufficient and in this case temporal propagation is needed. I find that the authors are using the state-of-the-art numerical techniques for the problem.
- Comment 2: the study on polarisation effects in fibers needs indeed to be put into context. The first observation of fiber baser cavity solitons are from 2009 from some of the authors, so I clearly disagree that these effects were observed in the 80s. We do not need to confuse conservative and dissipative literature and, in the latter case, driven solitons from laser solitons. All these systems have a deeply different physics. I think that, however, this point has been fairly raised and that the clarification of the authors allows to put their work into a better context now.
- Comment 3: I find this a rather minor point. That cavity solitons have been centrally studied as optical memories is undeniable (Barland et al. 'Cavity solitons as pixels in semiconductor microcavities' Nature volume 419, pages699–702(2002)), there is almost 30 years of literature. That temporal cavity solitons are more useful for metrology applications in microcombs than as memories is, indeed, in my view, a matter of opinion, as Referee 2 also concedes. Given this comment, I think that the authors have well explained that , because the basic physics is the same, this work provides critical advancement for both fields, so I would say that also this concern has been cleared.

Referee 4: general response

We are very grateful for the Referee's positive assessment of our work. They write that the scientific innovation of our work is well suited for Nature Communications, that our experimental results are of very high quality and novel, and that the phenomenon explored in our work "has never been observed before". They also emphasise that all of the comments previously raised by Referee 2 have been addressed. The Referee presents no negative comments, and so we would like to close by simply thanking them again.

Reviewers' Comments:

Reviewer #3:

Remarks to the Author:

1. The authors have provided detailed and comprehensive responses to the points raised in my previous reports, and cleared reasonable doubts that the analysis captures the essence of the experimental observations.
2. There is no point in arguing the question of impact and novelty any further: Prior work and its relation to the present work has been discussed at length by the authors and the referees, enabling the editors to form their opinion on this question.

Referee 3: comment 1

The authors have provided detailed and comprehensive responses to the points raised in my previous reports, and cleared reasonable doubts that the analysis captures the essence of the experimental observations.

Response 1

We thank the Referee for carefully reading our revised manuscript and response. We are glad that they feel all the points raised have been cleared.

Referee 3: comment 2

There is no point in arguing the question of impact and novelty any further: Prior work and its relation to the present work has been discussed at length by the authors and the referees, enabling the editors to form their opinion on this question.

Response 2

We agree fully, yet remark that we have valued the scientific dialogue, and believe it has been helpful in optimising our manuscript.